# Source apportionment of potentially toxic elements in soils using APCS/MLR, PMF and geostatistics in a typical industrial and mining city in Eastern China

Cao Jianfei[1], Li Chunfang[1], Zhang Lixia[2], Wu Quanyuan[1]*, Lv Jianshu[1]

**1** College of Geography and Environment, Shandong Normal University, Ji'nan, China, **2** General Station of Geological Environment Monitoring of Shandong province, Ji'nan, China

☯ These authors contributed equally to this work.
\* wqy6420582@163.com

## Abstract

Source apportionment of potentially toxic elements in soils is a critical step for devising soil sustainable management strategies. However, misjudgment or imprecision can occur when traditional statistical methods are applied to identify and apportion the sources. The main objective of the study was to develop a robust approach composed of the absolute principal component score/multiple linear regression (APCS/MLR) receptor model, positive matrix factorization (PMF) receptor model and geostatistics to identify and apportion sources of soil potentially toxic elements in typical industrial and mining city, eastern China. APCS/MLR and PMF were applied to provide robust factors with contribution rates. The geostatistics coupled with the variography and kriging methods was used to present factors derived from these two receptor models. The results indicated that mean concentrations of As, Cd, Cr, Cu, Hg, Ni, Pb and Zn exceeded the local background levels. Based on multivariate receptor models and geostatistics, we determined four sources of eight potentially toxic elements including natural source (parent material), agricultural practices, pollutant emissions (industrial, mining and traffic) and the atmospheric deposition of coal combustion, which accounted for 68%, 12%, 12% and 9% of the observed potentially toxic element concentrations, respectively. This study provides a reliable and robust approach for potentially toxic elements source apportionment in this particular industrial and mining city with a clear potential for future application in other regions.

## Introduction

In recent years, soil potentially toxic element pollution has been a worldwide environmental problem and has attracted much attention due to cumulative toxicity and persistence [1–3]. Especially in the eastern coastal areas of China, intensive human activities have led to the enrichment of potentially toxic elements in soils, threatening food safety and human health [4]. The concentration of soil potentially toxic elements is subjected to natural background

**Data Availability Statement:** All relevant data are within the manuscript and its Supporting Information files.

**Funding:** This study was supported by the National Natural Science Foundation of China (No. 41371395), the Natural Science Foundation of Shandong Province (ZR2017BD011), the China Postdoctoral Science Foundation (2017M622256) and the Key Technology Research and Development Program of Shandong (2017CXGC304, 2019GSF109034). The founders play any role in the study design, data collection and analysis, decision to publish, or preparation of the manuscript.

**Competing interests:** The authors declare that they have no conflicts of interest to disclose.

levels and human inputs [5]. The former arises from the weathering of geological parent rocks [1, 6]. Human input pathways include mining, waste disposal, sewage irrigation, vehicle exhaust emissions, atmospheric deposition, fertilizer and pesticide application, among other human activities [7, 8].

Source apportionment can contribute to determining the enrichment of potentially toxic elements from natural sources and complex human activities, and identify the contribution rate of each source. This analysis is crucial for devising soil sustainable management strategies so as to prevent or reduce potentially toxic element pollution. Among the methods involved in source apportionment, qualitative and quantitative analyses are commonly used. Multivariate statistical analyses belong to the former, i.e., principal component analysis (PCA) and factor analysis (FA), which have been widely used to assess pollution status and identify the sources of potentially toxic elements in soils [9–12]. These methods can determine the most significant factors by reducing dimensions and explaining potential sources of pollution. However, quantitative analysis cannot be achieved with the above methods. In this case, receptor models have been applied to quantify the sources of soil potentially toxic elements, i.e., chemical mass balance (CMB), absolute principal component score/multiple linear regression (APCS/MLR) and positive matrix factorization (PMF) [13–16]. CMB is a basic receptor model and requires both the concentration of potentially toxic elements and the input of source profiles. APCS/MLR and PMF are more efficient than CMB because they do not require source profiles [17, 18]. APCS/MLR evolved from PCA, and source contributions are obtained through carrying out the regressions between potentially toxic element contents and APCS. PMF uses experimental uncertainties in the data matrix and decomposes a data matrix into factor contributions and factor profiles under the non-negative constraint [15]. Due to theoretical differences, the results from receptor models differ, and each variable represents a source. To provide robust factors and better interpret the sources, previous studies have commonly applied multiple receptor models simultaneously based on the same datasets [19, 20]. These factor analysis methods still contain shortcomings, e.g. explore pollution sources based on previous knowledge, which may result in misjudgment or imprecision.

The spatial correlations between sampling points contain important information for interpret the potential soil potentially toxic element pollution source. There are two main groups of interpolation techniques: deterministic (polynomial, inverse distance weighted, and radial) and geostatistical (ordinary kriging, simple kriging, universal kriging, probability kriging, indicator kriging and disjunctive kriging) [21]. Because the geostatistical method coupled with the variography and kriging methods could quantify the spatial autocorrelation among measured points and account for the spatial configuration of the sample points around the prediction location, they have been widely used to provide insights into the spatial correlations of soil properties [22–26]. The spatial continuous variations, including structural spatial variations and random spatial variations, are calculated in the variogram. Kriged maps can characterize the hotspots and outlines. Previous studies have successfully used the spatial variation and spatial distribution of potentially toxic elements in soils to identify risk areas, which have been superimposed with land use maps to predict potentially toxic element pollution sources [27–29]. However, prior studies have rarely explored the spatial variation information of factor variables and the superposition information of kriged maps that form factor variables and potentially toxic elements, which contain important information for source apportionment. Therefore, this approach is expected to effectively integrate the multivariate receptor models and geostatistics for source apportionment and reduce the misjudgment and inaccuracy error.

In our study, we chose the northern plain of Longkou in Eastern China, as a typical region, where human activities are intensive due to the rapid development of industry and mining. Potentially toxic elements of As, Cd, Cr, Cu, Hg, Ni, Pb and Zn in 138 surface soil samples

were collected [30]. Based on the proposed approach composed of APCS/MLR, PMF and geostatistics, our specific objectives were to (1) provide robust source factors with contribution rates using multivariate receptor models, including APCS/MLR and PMF, (2) apply geostatistics to present those source factors to provide more objective and useful information in source apportionment, and (3) identify and apportion sources of soil potentially toxic elements in typical industrial and mining cities.

## Material and methods

### Study area

Longkou is a typical industrial and mining city in eastern China [31]. The research was conducted in the northern plain of Longkou City (37°34'35"N—37°44'49"N, 120°13'4"E—120°40'47"E), which covers an area of 500 km$^2$ (Fig 1). The study areas are priority areas in which industries have rapidly developed under the support of state policies. The multitude of emission sources have made this a typical area for verifying source apportionment models [32]. The area is characterized by mineral resource exploitation, including coal, gold mine and lead zinc mining, and the abundant natural resources have promoted the development of preliminary industrial enterprises, such as iron-making plants, paper mills and electroplating factories [33]. There is approximately 300 km$^2$ of agricultural land in this area, mainly wheat and maize planting in the west, apple and grape orchards in the east, and vegetable planting areas in the north (Fig 2). This study area has a temperate monsoon climate with an average annual temperature of 12°C and a mean annual precipitation of 600 mm. Parent material are composed of marine sediments in the western of study area, bordered by alluvium and moorstone in the southeastern region (Fig 3). A water shortage problem has emerged in Longkou [31]. Untreated water from industrial activities was used to irrigate farmland over the period of a decade until a sewage disposal apparatus was built in 2002; presently, agricultural activities use the disposed water [31].

No specific permissions were required for these locations and activities in the field sampling and we confirmed that the field studies did not involve any threat to endangered or protected species.

### Soil sampling and chemical analysis

A total of 138 soil samples were collected in the summer of 2017 based on the grid layout sample point method [27–29]. Sample sites were selected according to a sampling density of less than 2 km based on Landsat images, and each sample consisted of a mixture of five subsamples collected from five spots across an area of approximately 30 m$^2$. Each soil sampling site was first classified based on land use types including 33 for industrial and mining use, 34 for grain crop use, 24 for orchard use, 12 for vegetable use, 22 for residential use, and 13 near to roads. If the designed site was unavailable for sampling (such as if it contained a building), an alternative location was selected as close to the original as possible to find natural soils. All subsamples were collected at a depth of 0–20 cm using a stainless-steel shovel. At each sampling site, an approximate 1kg of the soil sample were mixed thoroughly in a polyethylene bag. After airdrying, the collected soils samples were sieved to 2 mm, and ground to powder that could pass through a 0.149-mm mesh for physical-chemical analysis. The geographical locations of the sampling points were recorded by a GPS receiver, as shown in Fig 1.

In the laboratory, soil pH values were measured by a pH meter in a 1:2.5 soil–water suspension, and organic matter (SOM) contents were analyzed using oil bath-$K_2CrO_7$ titration [34]. $HSO_4$-$HNO_3$-HF was used to digest the soil for analyzing the As, Cd, Cr, Cu, Hg, Ni, Pb and Zn contents. The Cr, Cu, Ni, Pb and Zn contents were analyzed using a flame atomic

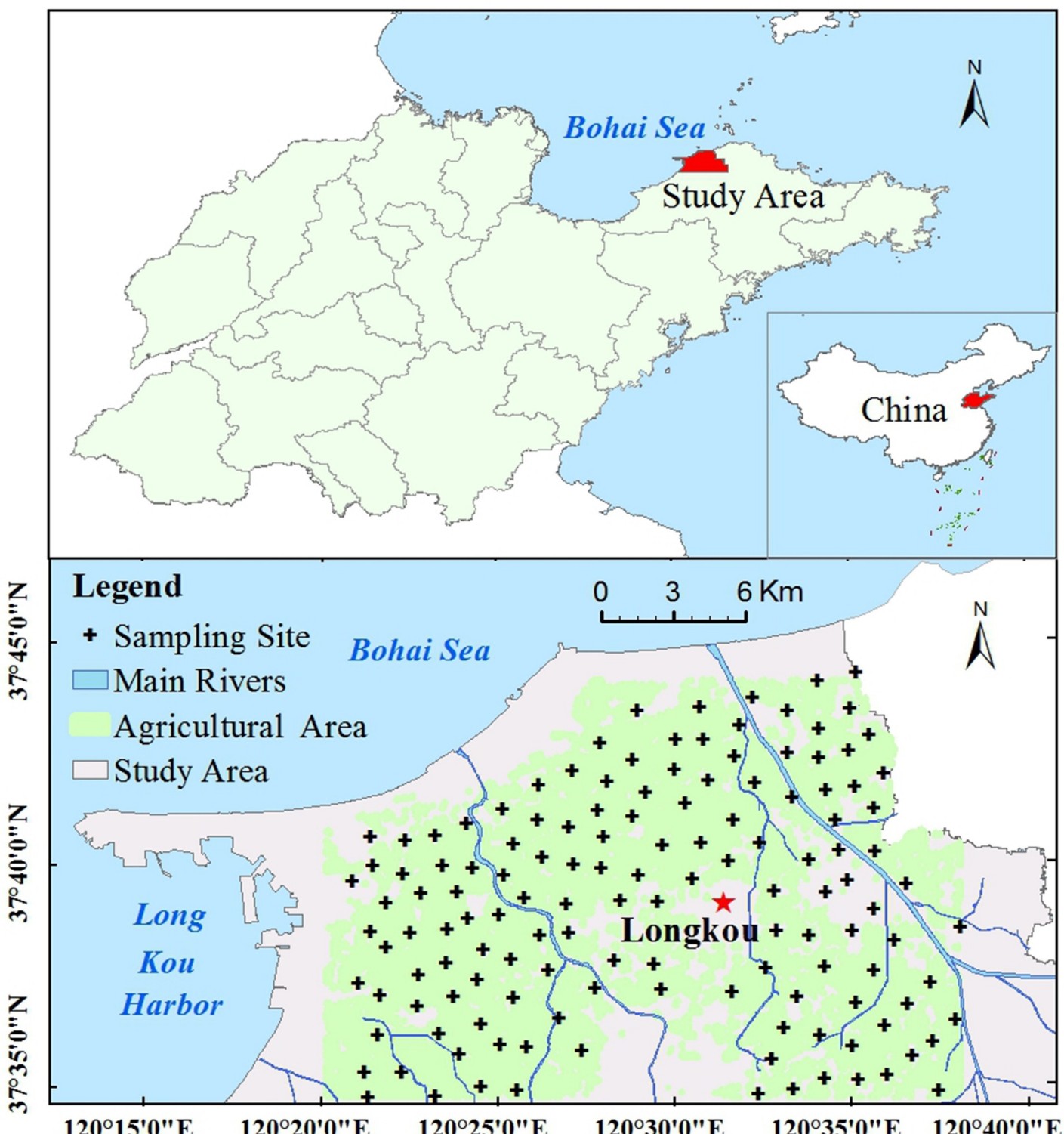

**Fig 1. Geographical location of the study area with sampling sites.** (The map was generated using free, open access data sources from the National Geomatics Center of China).

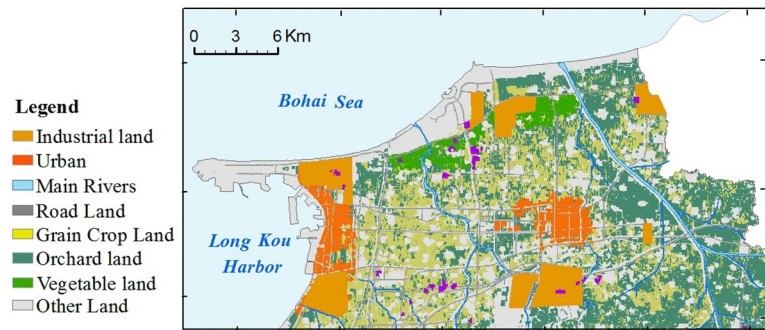

**Fig 2. The maps of land use types in the study area.** (The land used map was generated by interpreting free Landsat images).

absorption spectrophotometer (240 AA Agilent, USA), Cd contents were determined using a graphite furnace atomic absorption spectrophotometer (AA-7000 Shimadzu, Japan), and As and Hg concentrations were determined with an atomic fluorescence spectrophotometer (AFS230E Haiguang Analytical Instrument Co., Beijing, China). For details on the measurements, please refer to the related literature [34]. A standard reference material obtained from the Center for National Standard Reference Material of China (http://www.biobw.com/), was used for quality control. The recovery rate and standard reference material were examined under strict monitoring, and the chemical analysis process followed the standard for geochemical evaluation of land quality (DZ/T0295–2016) in China. The limit of recovery was 94% ~106%.

## Source apportionment method

The source apportionment method framework is show in Fig 4, which integrates APCS/MLR, PMF and geostatistics. APCS/MLR and PMF were simultaneously applied to the potentially toxic element concentration dataset to provide more factors with contribution rates. The geostatistics were applied to present those factors. The spatial variant structure of those factor variables was used to preliminarily determine which factors belonged to natural sources or anthropogenic sources. The spatial distribution characteristics of the factors and eight potentially toxic elements were mapped via ordinary kriging and were superposed on the auxiliary environmental data (such as land use types and parent materials) to locate the potential sources.

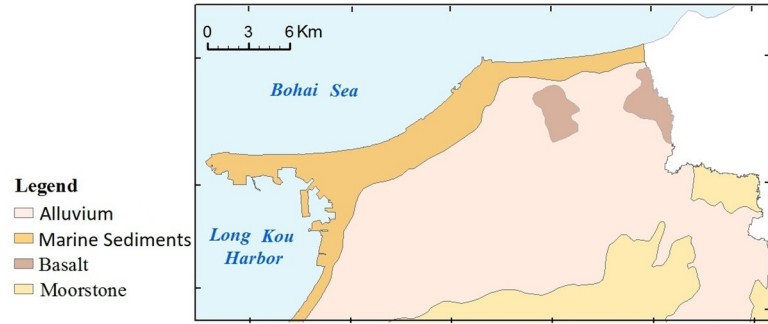

**Fig 3. The maps of parent materials in the study area.** (The parent materials map was generated using free, open access data sources from the Geological survey development research center of China).

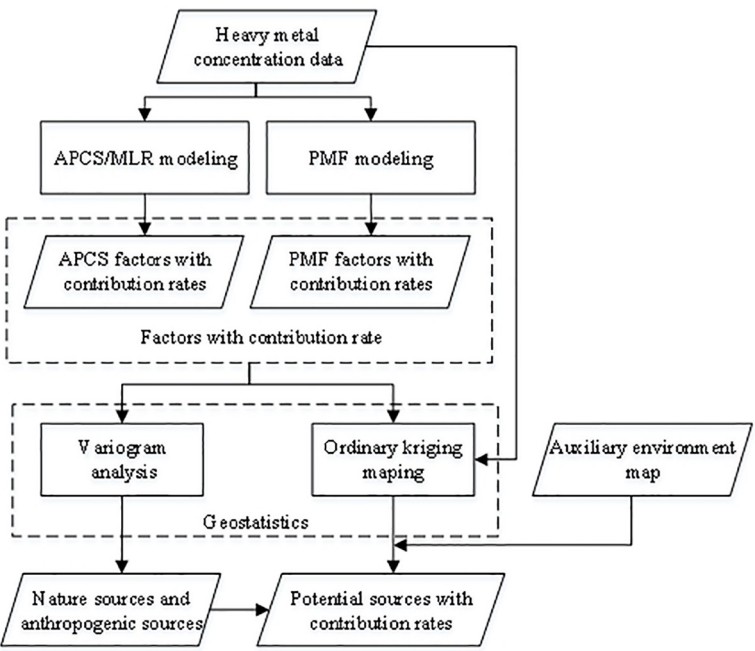

**Fig 4. Source apportionment method.**

**APCS-MLR receptor model.** The APCS-MLR receptor model applies two mathematical methods, i.e., a combination of a multiple linear regression model (MLR) and the absolute principal component scores (APCS) [35]. This model was calculated using SPSS 22.0 software (IBM Inc., USA). The first procedure normalizes the raw data as follows:

$$Z_{ij} = \frac{C_{ij} - \bar{C}}{\sigma_i} \tag{1}$$

where $Z_{ij}$ is the content after normalization, $C_{ij}$ is the content of the $i$th sample of the $j$th element, and $C_j$ and $\sigma_j$ represent the respective average content and standard deviation of the $j$th element, respectively.

Then, a comparison sample $(Z_0)_i$ with a content of 0 was inserted, and normalization was conducted as follows:

$$(Z_0)_i = \frac{0 - \bar{C}_i}{\sigma_i} = -\frac{\bar{C}_i}{\sigma_i} \tag{2}$$

The APCS for the factors are estimated by subtracting the factor scores of $Z_0$ from the factor scores of true samples. The apportionment to $C_j$ can be evaluated via MLR as follows:

$$C_i = b_{0i} + \sum_{p=1}^{n} (APCS_p \bullet b_{pi}) \tag{3}$$

where $b_{0i}$ is the constant term in the MLR and $b_{pi}$ is the regression coefficient for the $p$th source of the $i$th element. The adjusted score of the $p$th factor is $APCS_p$, and the average contribution of the $p$th source to $C_i$ can be interpreted as $APCS_p \bullet b_{pi}$.

**Positive matrix factorization model.** PMF is a method that decomposes the elemental content matrix into a factor contribution matrix and a factor component spectrum matrix [36]

and is performed with the US-EPA PMF 5.0 model. First, the original elemental matrix X nm with the order n*m can be described as

$$E_{nm} = X_{nm} - \sum_{j=1}^{p} G_{np}F_{pm} \tag{4}$$

where G(n*p) and F(p*m) represent the matrices of the factor contribution and factor profile, respectively, and E(n*m) is the matrix of the residual error.

Furthermore, the objective function Q is the diagnostic index of model performance, and the Q value from the model result must be close to the reference value. Q can be expressed as follows:

$$Q = \sum_{i=1}^{m}\sum_{j=1}^{n} (E_{ij}/\sigma_{ij})^2 \tag{5}$$

where $E_{ij}$ is the residual error of the $i$th element of the $j$th sample, $\sigma_{ij}$ is the uncertainty of the $i$th element of the $j$th sample, and all values in the above calculation process are dimensionless. Finally, the uncertainty (U) is determined using the EPA PMF 5.0 User Guide (U.S. Environmental Protection Agency, 2014). If all elemental contents are greater than the method detection limit (MDL), the uncertainty calculation is performed as follows:

$$U = \sqrt{(ErrorFraction \times concentration)^2 + (0.5 \times MDL)^2} \tag{6}$$

**Geostatistical method.** Geostatistics was used to analyze the spatial correlation of the APCS, PMF-factors and eight potentially toxic elements and to minimize the estimation error in source identification. Three structure variance theoretical models of three structures (spherical, Gaussian and exponential) were employed to measure the degree of spatial variability. The determination coefficient ($R^2$) and residual sum of squares (RSS) were used to evaluate the optimal structure variation model. The nugget value ($C_0$), sill value ($C_0+C$) and variable range (A) were the main parameter of variation model. The nugget effect ($C_0/C_0+C$) was used to distinguish between regional factors (natural factors) and nonregional factors (human factors) for heavy metal enrichments. There are three classes for the $C_0/C_0+C$ values, strong spatial autocorrelation ($C_0/C_0+C \leq 0.25$), moderate spatial autocorrelation ($0.25 < C_0/C_0+C < 0.75$), and weak spatial autocorrelation ($0.75 \leq C_0/C_0+C$) [37]. The variable range (A) represents the range of spatial autocorrelation under a certain observation scale. The estimation process of the structure variance model was performed with GS+ 7.0 (R Development Core Team). Ordinary kriging (OK) was used for interpolation and characterizing hotspots and outlines of hazardous areas, which was implemented using ArcGIS 10.1 (ESRI Inc., USA).

## Results

### Description of potentially toxic elements

The descriptive statistics of the soil potentially toxic element contents in the study area are shown in Table 1. The mean soil pH value ranged from 6.26 to 7.88, with a mean value of 7.0. The soil organic matter (SOM) content ranged from 5.46 g kg$^{-1}$ to 42.22 g kg$^{-1}$, with a mean value of 24.94 g kg$^{-1}$, and these values were higher than the background values [38]. Overall, the average potentially toxic element contents in all samples were below level II of the Environmental Quality Standard for Soils (EQSS) of China [39] but exceeded the corresponding background values [38]. In particular, Cd, Cu and Hg were 1.81, 1.80 and 1.63 times higher than the background values, respectively, suggesting that the topsoil, which is affected by human

**Table 1. Descriptive statistics of potentially toxic elements in the study area (n = 138, units in mg kg$^{-1}$).**

| Species | Min | Max | median | percentile 25 | percentile 75 | Mean | S.D. | C.V.(%) | Skewness | Background | I$_{geo}$ | EQSS |
|---|---|---|---|---|---|---|---|---|---|---|---|---|
| As | 3.63 | 10.26 | 7.98 | 6.73 | 8.97 | 7.96 | 1.47 | 18.51 | 0.34 | 6.30 | -0.25 | 30 |
| Cd | 0.049 | 0.42 | 0.23 | 0.13 | 0.37 | 0.20 | 0.11 | 56.00 | 9.91 | 0.11 | 0.28 | 0.30 |
| Cr | 46.7 | 96.83 | 61.08 | 50.10 | 64.8 | 61.10 | 10.58 | 17.32 | 0.23 | 56.20 | -0.46 | 200 |
| Cu | 20.34 | 50.86 | 35.35 | 21.73 | 43.78 | 35.30 | 16.60 | 47.02 | 6.61 | 19.60 | 0.26 | 100 |
| Hg | 0.025 | 0.064 | 0.048 | 0.038 | 0.057 | 0.049 | 0.02 | 42.86 | 2.55 | 0.03 | 0.12 | 0.50 |
| Ni | 19.18 | 47.6 | 26.62 | 22.91 | 31.79 | 26.59 | 5.26 | 19.78 | 0.38 | 23.50 | -0.41 | 50 |
| Pb | 15.75 | 63.92 | 35.13 | 28.90 | 50.36 | 35.08 | 13.37 | 38.11 | 8.20 | 25.40 | -0.12 | 300 |
| Zn | 53.99 | 104.32 | 77.85 | 60.11 | 91.57 | 77.89 | 14.05 | 18.04 | 1.02 | 56.10 | -0.11 | 250 |
| PH | 6.26 | 7.88 | 7.00 | 6.68 | 7.38 | 7.00 | 0.35 | 5.00 | 0.37 | - | - | - |
| SOM (g kg$^{-1}$) | 5.46 | 42.22 | 14.99 | 8.86. | 18.25 | 14.94 | 5.04 | 33.73 | 1.44 | 13.00 | - | - |

activities, was enriched by these potentially toxic elements. Compared with the surrounding cities with developed industry and mining, such as Rizhao [27], Guangrao [28] and Ju County [29], it was found that the average value of potentially toxic elements in soils had the above similar characteristics as in Longkou, and Cd and Hg were also considered to be the most risky. To further evaluate the enrichment degree of potentially toxic elements, the index of geo-accumulation (I$_{geo}$) was calculated using Muller's equation [40], which indicated that the soil ranged from not contaminated to moderately contaminated with respect to Cd, Cu, and Hg, which were ranked as Cd>Hg>Cu, and the soil was not contaminated with the other elements.

The coefficient of variation (C.V.) is a dimensionless expression of the standard deviation and can better reflect fluctuations in potentially toxic element contents [41]. The highest C.V. was found for Pb followed by Cr and Cd, which indicates high variations of these metals in the soil, and exhibited the following order: Cd > Cu > Hg > Pb > Ni > As > Zn > Cr. The skewness of the studied potentially toxic elements exhibited the following order: Cd > Pb > Cu > Hg > Zn>Ni>As>Cr. Overall, Cd, Pb, Cu, Hg and Zn were found to be higher than one which indicates right handed skewness. It suggest that these soil metals may be affected by human factors [42].

## Source factors of soil potentially toxic elements

The potentially toxic element contents in the soil samples were analyzed by PCA (Table 2). The first four factors were extracted, which explained 79.60% of the total variance. The first factor (F1) accounted for 27.15% of the total variance and showed strongly positive loadings of Cr and Ni and a moderate loading of As. F2 explained approximately 18.94% of the total

**Table 2. Factors loadings of potentially toxic elements in soils.**

| | F1 | F2 | F3 | F4 |
|---|---|---|---|---|
| Cr | 0.923 | -0.075 | 0.168 | -0.004 |
| Cu | -0.103 | 0.894 | 0.06 | -0.098 |
| Ni | 0.916 | -0.055 | 0.036 | -0.156 |
| Pb | 0.137 | -0.005 | 0.925 | 0.139 |
| Zn | 0.137 | 0.772 | 0.002 | 0.272 |
| Cd | 0.267 | 0.123 | 0.683 | -0.483 |
| As | 0.67 | 0.288 | 0.269 | 0.188 |
| Hg | 0.005 | 0.13 | 0.019 | 0.924 |
| Variance contribution rate/% | 27.145 | 18.938 | 17.635 | 15.876 |
| Accumulated Variance Contribution Rate/% | 27.145 | 46.083 | 63.718 | 79.594 |

**Table 3. Contribution rate of each factor to potentially toxic elements derived from APCS/MLR and PMF.**

|  | APCS/MLR | | | | | | | PMF | | | | | |
|  | Ratio | $R^2$ | F1 | F2 | F3 | F4 | Unidentified | Ratio | $R^2$ | F1 | F2 | F3 | F4 |
|---|---|---|---|---|---|---|---|---|---|---|---|---|---|
| As | 0.91 | 0.60 | 0.71 | 0.13 | 0.08 | 0.20 | -0.12 | 1.06 | 0.81 | 0.81 | 0.07 | 0.05 | 0.07 |
| Cd | 1.00 | 0.78 | 0.28 | 0.13 | 0.57 | -0.13 | 0.15 | 1.05 | 0.96 | 0.36 | 0.14 | 0.46 | 0.04 |
| Cr | 1.03 | 0.87 | 0.98 | -0.09 | 0.18 | -0.01 | -0.06 | 0.93 | 0.60 | 0.94 | 0.02 | 0.02 | 0.02 |
| Cu | 1.14 | 0.82 | 0.67 | 0.29 | 0.06 | -0.10 | 0.08 | 0.97 | 0.86 | 0.57 | 0.32 | 0.04 | 0.07 |
| Hg | 0.94 | 0.87 | 0.58 | 0.13 | 0.02 | 0.39 | -0.12 | 1.07 | 0.75 | 0.60 | 0.01 | 0.01 | 0.38 |
| Ni | 1.07 | 0.86 | 0.96 | -0.05 | 0.04 | -0.06 | 0.11 | 1.01 | 0.98 | 0.92 | 0.02 | 0.03 | 0.03 |
| Pb | 0.96 | 0.89 | 0.55 | -0.01 | 0.34 | 0.14 | -0.02 | 1.12 | 0.51 | 0.57 | 0.1 | 0.27 | 0.06 |
| Zn | 1.11 | 0.68 | 0.63 | 0.38 | 0.01 | 0.08 | -0.10 | 0.98 | 0.73 | 0.63 | 0.28 | 0.07 | 0.02 |
| Mean | - | - | 0.67 | 0.11 | 0.16 | 0.06 | -0.01 | - | - | 0.68 | 0.12 | 0.12 | 0.09 |

variation and had a highly positive loading of Cu and a moderate loading of Zn. F3 explained 17.64% of the total variance and had a highly positive loading of Pb and a moderate loading of Cd. F4 accounted for 15.88% of the total variance and a highly positive loading of Hg.

The contributions of different factors were calculated using the APCS/MLR receptor model (Table 3). The accuracy of APCS/MLR was assessed via the $R^2$ and predicted/observed values. The $R^2$ parameters varied between 0.60 and 0.89, and the predicted/observed values ranged from 0.91 to 1.14, indicating that the APCS-MLR model had high accuracy. F1 primarily contributed to As, Cr and Ni with values of 71%, 98% and 96%, respectively. F2 dominated Cu (29%) and Zn (38%). F3 explained the 57% of Cd and 34% of Pb variations, and the six other potentially toxic elements had positive values. F4 explained 39% of Hg. However, the APCS exhibited a component that was not accounted for, i.e., the intercepts of the regressions, which ranged from -12% to 15%. The mean of the eight potentially toxic elements represented four factors, and the contributions of the four factors to potentially toxic element pollution in the study area were 67%, 11%, 16% and 6%.

In the PMF model, the number of optimal factors was determined to be four through training experiments, which is consistent with the APCS/MLR results. The results of the PMF model are shown in Table 4, and the Q (robust) value was 12723.8. All potentially toxic elements in the PMF model had a high correlation, with $R^2 > 0.51$ and $0.93 <$ Ratio (Predicted/Observed) $< 1.12$. Cr, Cu, Hg, Ni and Zn had the highest correlations in F1, which dominated the contribution and presented values ranging from 51% to 94%. Cu had the highest concentration in F2 and accounted for 32%, and Zn represented 28% of the content related to F2. F3 influenced Cd (46%) and contributed to Pb with values of 28%. F4 had the strongest contribution to Hg (38%). The mean of the eight potentially toxic elements represented four factors, and the contributions of the four factors to potentially toxic element pollution in the study area were 68%, 12%, 12% and 9%. Overall, the grouping of potentially toxic elements from the APCS and PMF models were similar and exhibited comparable factor contribution rates.

## Spatial variant structures of factors and potentially toxic elements

To perform the statistical analysis more efficiently, the variables were log-transformed. After data transformation, their skewness were reduced and the variable distributions approximate the normal distribution. The different optimal variogram models of APCS, PMF-factors and potentially toxic elements are shown in Table 4. The RSS and $R^2$ of all the optimal variogram optimal models varied among 0.011 and 0.071, 0.576 and 0.812, indicating that the fitting results were satifactory. The $C_0/(C_0+C)$ values of APCS1, PMF-F1, As, Cr and Ni were less than 0.25, which showed a strong spatial auto correlation and may have been indicative of

**Table 4. Optimal variation function model of the source factors and soil potentially toxic elements.**

| | Model | Nugget($C_0$) | Sill ($C_0+C$) | Proportion $C_0/(C_0+C)$ | Range (A)/m | RSS | $R^2$ |
|---|---|---|---|---|---|---|---|
| Lg APCS1 | Exponential | 0.008 | 0.039 | 0.210 | 6790 | 0.024 | 0.620 |
| Lg APCS2 | Spherical | 0.051 | 0.105 | 0.486 | 7872 | 0.037 | 0.763 |
| Lg APCS3 | Exponential | 0.068 | 0.108 | 0.627 | 6930 | 0.071 | 0.728 |
| Lg APCS4 | Exponential | 0.048 | 0.106 | 0.453 | 5115 | 0.020 | 0.587 |
| Lg PMF-F1 | Exponential | 0.012 | 0.005 | 0.240 | 4643 | 0.020 | 0.568 |
| Lg PMF-F2 | Spherical | 0.042 | 0.074 | 0.569 | 9492 | 0.011 | 0.812 |
| Lg PMF-F3 | Exponential | 0.055 | 0.076 | 0.720 | 7800 | 0.042 | 0.617 |
| Lg PMF-F4 | Exponential | 0.041 | 0.082 | 0.500 | 5974 | 0.002 | 0.783 |
| Lg As | Exponential | 0.022 | 0.106 | 0.207 | 8768 | 0.013 | 0.728 |
| Lg Cd | Spherical | 0.073 | 0.105 | 0.700 | 7852 | 0.002 | 0.780 |
| Lg Cu | Spherical | 0.045 | 0.069 | 0.652 | 9796 | 0.011 | 0.642 |
| Lg Cr | Exponential | 0.033 | 0.282 | 0.117 | 5947 | 0.012 | 0.590 |
| Lg Hg | Exponential | 0.022 | 0.037 | 0.595 | 7567 | 0.056 | 0.683 |
| Lg Ni | Spherical | 0.011 | 0.085 | 0.129 | 4967 | 0.002 | 0.783 |
| Lg Pb | Exponential | 0.065 | 0.131 | 0.495 | 7200 | 0.042 | 0.647 |
| Lg Zn | Spherical | 0.032 | 0.098 | 0.327 | 6860 | 0.003 | 0.576 |

natural factors. Other variables, with $C_0/(C_0+C)$ values between 0.25 and 0.75, showed moderate spatial auto correlation and may represent human activity factors. The spatial variability of potentially toxic elements based on $C_0/(C_0+C)$ was similar to the statistical results presented in Section 3,1, which exhibit an order of Cd> Cu> Hg> Pb> Zn> Ni>As>Cr. The A of the variables ranged from 4630 m to 9796 m, which was larger than the actual sampling interval and better represented the spatial variant structure of the potentially toxic elements.

## Spatial distribution characteristics of factors and potentially toxic elements

The kriged maps of APCS, PMF-factors and potentially toxic elements are shown in Figs 5 and 6, which were used to delineate the hotspots and outlines of hazardous areas. In Fig 5, the spatial distributions of APCS were similar to the overall trend of the corresponding PMF-factor, but the second factor had local differences, in which the spatial distribution of APCS2 had one more hotspot than PMF-F2. The spatial distributions of potentially toxic elements and those factors were clearly correlated, and this association was consistent with the results of the receptor model analysis (in section 3.2). Furthermore, those kriged maps were superposed on the auxiliary environmental data (land use types and parent materials) to locate the potential sources. As, Cr and Ni exhibited similar spatial distribution patterns to APCS1 and PMF-F1 and were characterized by higher values in the southeastern region, which was similar to the parent materials (Fig 3). The outlines of higher values of APCS2, PMF-F2, Cu and Zn were in accordance with the types of farmland (Fig 2). The common distribution characteristics of APCS3 and Pb were similar to human activity areas, including mining districts, industrial areas and traffic areas. PMF-F3 and Cd had higher values in the southern part close to the urban and industrial areas (Fig 2). The higher values of APCS4, PMF-F4 and Hg covered most of the study areas.

## Discussion

### Identification of the potentially toxic element source in soils

**Source interpretation of the first factor.** Cr, Ni and As had high and positive loading values in F1 based on the APCS/MLR and PMF modeling results. The mean values of Cr, Ni, and

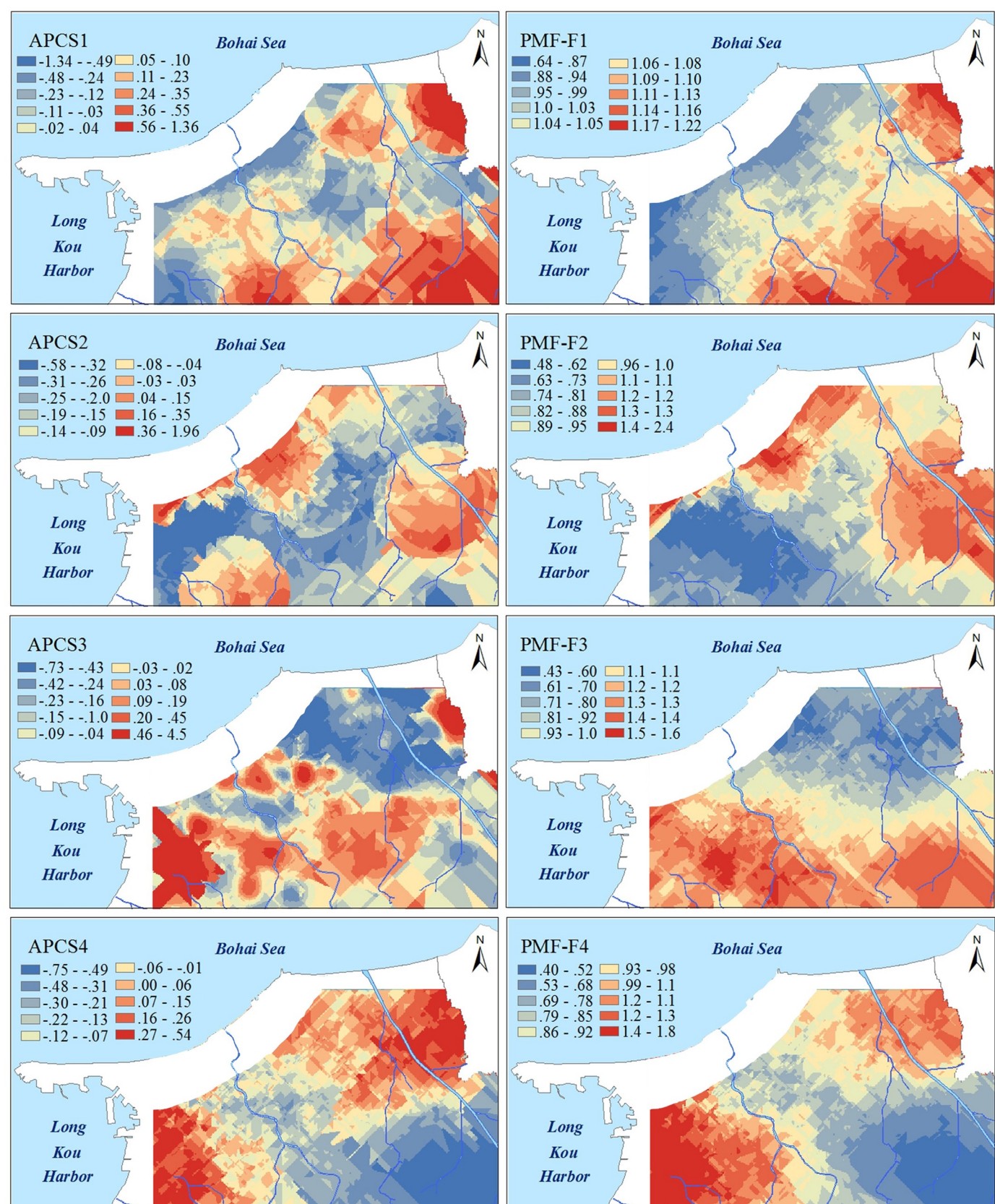

**Fig 5. Kriged interpolation of the APCS and PMF factors.** (The interpolated map plotted with the optimal ordinary kriged interpolation model).

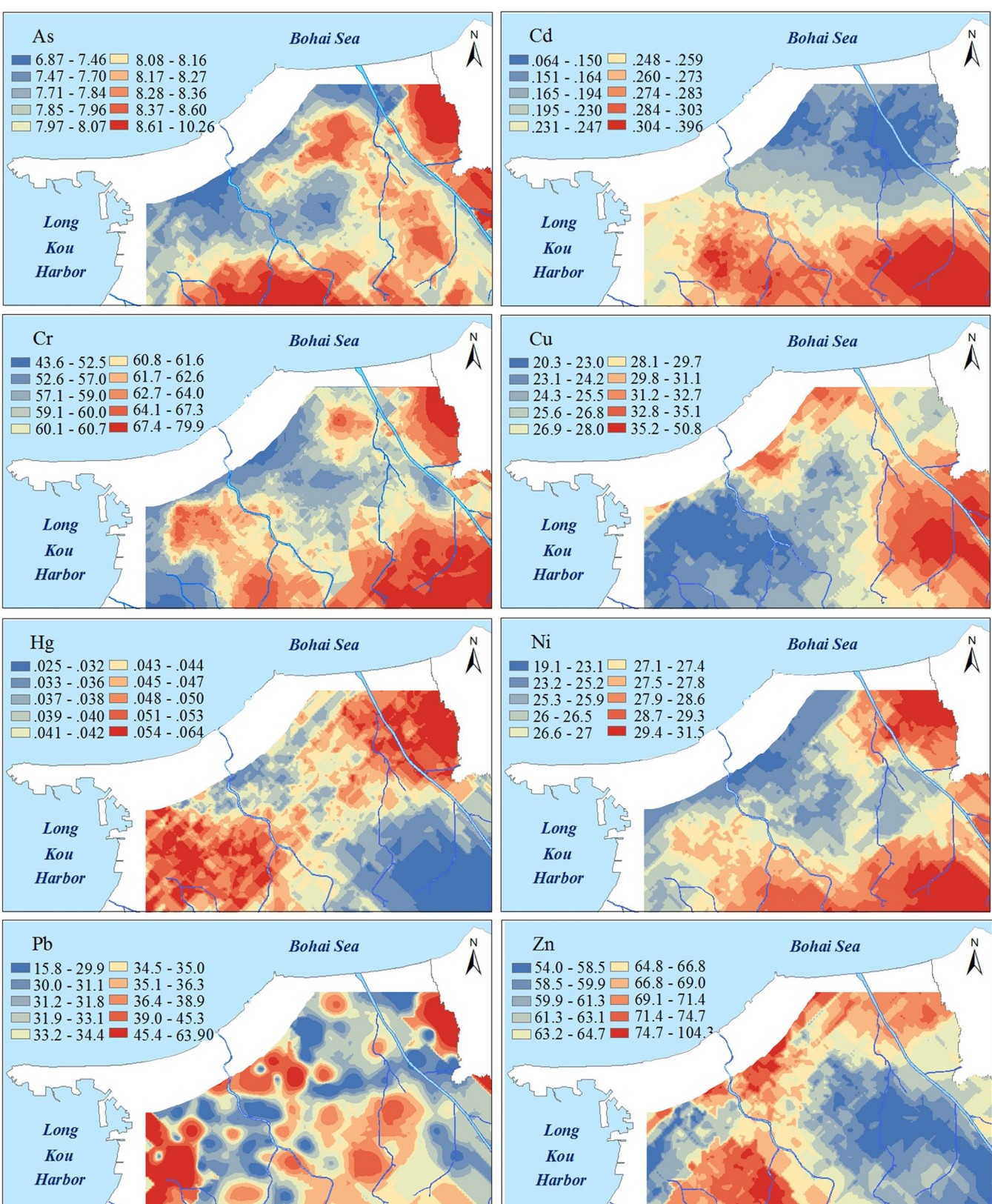

**Fig 6. Kriged interpolation of the concentration of potentially toxic elements.** (The interpolated map plotted with the optimal ordinary kriged interpolation model).

As were close to the respective background values, and the $I_{geo}$ value was negative, indicating that these three elements were less affected by human activities. F1 had a strong spatial autocorrelation that represented a natural factor [37, 43]. Martin et al. [44] studied potentially toxic element concentrations in topsoils in the Ebro basin and suggested that the grouping of Cr and Ni with other potentially toxic elements by multivariate analysis was generally regarded as the influence of natural source factors, which is consistent with the results of Nanos and Rodríguez Martin [45] for the Duero River basin, Lv et al. [27] for Ju County, Jiang et al. [46] for Changshu and Lv and Liu [6] for Boshan.

The results of the geostatistical analysis also show that the spatial patterns of F1 were consistent with the distribution of parent materials (Fig 5), with higher values in southeastern soils originating from granite. A soil with granite parent material, with a pH that is mainly neutral to acidic (Table 1), typically exhibits a poor toxic buffering capacity against potentially toxic element pollutants and is more likely to be enriched [47, 48]. Therefore, we confirmed that As Cr and Ni were classed into a lithological sources by the parent materials, and F1 represented a natural factor.

**Source interpretation of the second factor.**   Cu and Zn were highly related to F2 in the APCS/MLR and PMF results, and 68% and 38% of the respective variation was explained via PMF modeling (Table 3). The hotspots of APCS2, PMF-F2, Cu and Zn coincided with the spatial distribution of agricultural land types including grain crop land, orchard land and vegetable land. Agricultural chemical fertilizers are an important source of Cu and Zn enrichment, and phosphorus fertilizer is present in the highest amount in all inorganic fertilizers [49–52]. Cu and Zn are often used as additives in livestock diets to control scours [53].The amount of fertilizer applied to vegetable land is 5–10 times higher than that applied to other cultivated land [54]. Northern hotspots of F1 were shown to coincide with the vegetable boundary. A total of 1.9 thousand tons of chemical pesticides are used on the study area each year [55]. Cu-based fungicides ($CuSO_4 \cdot Cu(OH)_2 \cdot Ca(OH)_2 \cdot H_2O$) are widely used to control pests and diseases, and the eastern hotspots of Cu were consistent with orchard-growing areas.

The results from previous studies conducted by Jiang et al. [47] in Jiangsu, Guan et al. [56] in the Hexi Corridor and Hu et al. [3] in a peri-urban area of Nanjing also showed that enrichment of Cu and Zn in agricultural soil were mainly associated with livestock manure, chemical fertilizers and pesticides in agricultural soils. Therefore, we confirmed that F2 represented agricultural practices.

**Source interpretation of the third factor.**   F3 was strongly related to Pb and Cd and exhibited had high spatial variability (Tables 1 and 4). The hotspots of the F3 kriged maps were distributed in areas, including mining districts, industrial areas and urban areas. There are 65 heavy industrial enterprises and 115 light industrial enterprises in the region, including electric power plants, paper mills, and electroplating [55]. Cd is the main raw material in the electroplating industry because of its anticorrosive effect on acid and is also widely used in the production of dyes and power generation. The long-term production and operation of these industries will lead to the enrichment of Cd in the surrounding soil. There are seven mining areas in the study area with an annual coal output of 6.7 million tons [55]. The wastewater produced in the long-term mining process carries a certain amount of Pb into cropland [57]. Moreover, the combustion of petroleum and the use of catalysts in industrial production and transportation are the major sources of Pb [58, 59]. In summer, F3 represented industrial, mining and traffic emissions.

**Source interpretation of the fourth factor.**   F4 was dominated exclusively by Hg in the APCS/MLR and PMF modeling results. Most of the values of the kriged map were higher than the background values, suggesting that the enrichment of Hg was related to atmospheric deposition [1, 60]. Moreover, the distribution of hotspot areas was similar to that of industrial land.

Many researchers have noted that coal burning is the most important source of Hg [61, 62]. Due to its high volatility, Hg rapidly migrates in gaseous and granular forms through dry and wet deposition [62–65]. In the study area, most of the power for industrial activities comes from coal combustion and oil burning, and energy-intensive industries such as the metallurgical and chemical industry account for 70% of industrial production [55]. Long-term industrial production led to the migration of Hg through exhaust emissions, thus resulting in the enrichment of Hg in soil. Therefore, we confirmed that F4 represented atmospheric deposition of coal combustion.

### Source contributions of soil potentially toxic elements

The source contributions of potentially toxic elements are presented in Table 3. In total, the solutions of the APCS/MLR results explained approximately 101% of all the sources, and PMF resolved 100% of the sources contributing to potentially toxic elements. In the APCS/MLR results, 67% of the potentially toxic elements originated from soil parent material, 11% of the potentially toxic elements originated agricultural practices, 16% of the potentially toxic elements originated from industrial, mining and traffic emissions, and only 6% of the total potentially toxic element contents were attributed to atmospheric deposition. In the PMF results, the largest source was from soil parent material (68%), followed by industrial, mining and traffic emissions (12%), agricultural practices (12%) and atmospheric deposition (9%). By comparing the results of the two models, the difference in the source contributions ranged from 1% to 4%, indicating that the source apportionment results were robust.

Regarding the potentially toxic elements, PMF provided more rational source contributions than the APCS/MLR results because APCS/MLR had negative and unidentified contributions. Based on the PMF modeling, As, Cr and Ni were mainly affected by soil parent material with contributions greater than 81%. Cu and Zn were dominated by soil parent material with contributions of 57.0% and 63%, respectively, but agricultural practices also accounted for 32% of Cu and 28% of Zn. Cd was mainly explained by soil industrial, mining and traffic emissions, with a value of 46% and soil parent material also accounted for 36% of Cd. Pb was controlled by soil parent material and industrial, mining and traffic emissions, with values of 57% and 27%. The Zn concentration (63%) was associated with parent materials, and it was also influenced by agricultural practices (28%). Hg was explained by soil parent material and atmospheric deposition from coal combustion, with values of 60% and 38%.

At the Chinese scale, Hg dramatically declined due to strict control of atmospheric Hg emission in China since 2010 [66]. Moreover, inputs of all the heavy metals from fertilizers decreased, because of the stricter fertilizer management and modernized fertilizer production technologies [66, 67]. Heavy metals are more likely to be enriched from fertilizers and sewage irrigation sources in North China, where water is scarce, than in the South China [67]. In provinces with high GDP (Guangdong, Jiangsu, Henan, and Shandong provinces), industrial and traffic activities sources contributed more heavy metals in soil, mainly Pb and Cd [68, 69]. In general, the source of soil potentially toxic element in the study area is similar to that in China as a whole. The law of the People's Republic of China on the prevention and control of soil pollution was came into effect on January 1, 2019, it constitute a new comprehensive control system of soil pollution. In the future, local governments can develop more effective soil pollution control strategies based on studies of the quantitative sources of potentially toxic element.

## Conclusions

This study provides a reliable and robust approach for potentially toxic elements source apportionment in this particular industrial and mining city with a clear potential. A robust approach

composed of APCS/MLR and PMF with geostatistics was proposed to identify the apportion sources of soil potentially toxic elements in the typical industrial and mining city of Longkou in Eastern China.

According to the local background levels of potentially toxic element contents, Cr, Cu, Ni, Pb, Zn, Cd, As and Hg had different levels of accumulation. Based on different theoretical foundations, APCS/MLR and PMF provided similar four factors. The representation of the results derived from the multivariate receptor models by geostatistics made the source apportionment analysis more robust and accurate because the representation information was correlated with the auxiliary environment data (land use types and parent materials). Spatial variation analysis revealed that the first factor was dominated by natural sources and the other factors were affected by anthropogenic sources. The spatial distribution of the factors and potentially toxic elements located the potential sources, including the natural sources caused by parent materials, agricultural practices, pollutant emissions (industrial, mining and traffic) and atmosphere deposition of coal combustion. Although PMF and APCS/MLR had similar source contributions for potentially toxic elements, PMF with positive values was more precise for source apportionment than APCS/MLR. Based on PMF, a total of eight potentially toxic elements explained 68%, 12%, 12% and 9% of the observed potentially toxic element concentrations. In addition, this research idea can be applied to other research areas.

## Supporting information

**S1 Table. The concentration of heavy metals in 138 topsoil samples.**
(XLSX)

## Author Contributions

**Conceptualization:** Cao Jianfei, Wu Quanyuan.

**Data curation:** Li Chunfang, Zhang Lixia.

**Formal analysis:** Cao Jianfei.

**Funding acquisition:** Li Chunfang, Wu Quanyuan.

**Investigation:** Li Chunfang, Zhang Lixia.

**Methodology:** Cao Jianfei.

**Resources:** Li Chunfang.

**Software:** Li Chunfang.

**Supervision:** Wu Quanyuan, Lv Jianshu.

**Validation:** Cao Jianfei, Li Chunfang.

**Visualization:** Cao Jianfei, Li Chunfang.

**Writing – original draft:** Cao Jianfei.

**Writing – review & editing:** Lv Jianshu.

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
