## [Decision Letter · Decision Letter 0]

3 Apr 2020

PONE-D-20-05425

Source apportionment of heavy metals in soils using APCS/MLR, PMF and geostatistics in a typical industrial and mining city in Eastern China

PLOS ONE

Dear mr Jianfei,

Thank you for submitting your manuscript to PLOS ONE. After careful consideration, we feel that it has merit but does not fully meet PLOS ONE’s publication criteria as it currently stands. Therefore, we invite you to submit a revised version of the manuscript that addresses the points raised during the review process.

We would appreciate receiving your revised manuscript by May 18 2020 11:59PM. To enhance the reproducibility of your results, we recommend that if applicable you deposit your laboratory protocols in protocols.io, where a protocol can be assigned its own identifier (DOI) such that it can be cited independently in the future. For instructions see: http://journals.plos.org/plosone/s/submission-guidelines#loc-laboratory-protocols

We look forward to receiving your revised manuscript.

Kind regards,

Andrés Rodríguez-Seijo, PhD

Academic Editor

PLOS ONE

Journal Requirements:

1. PLOS specifies that experiments, statistics, and other analyses are performed to a high technical standard; sample sizes are large enough to produce robust results; and methods are described in sufficient detail to allow another researcher to reproduce the experiment (http://journals.plos.org/plosone/s/criteria-for-publication#loc-3). We feel that your methods section did not include sufficient detail, including the manufacturer of all chemicals and instruments. Additionally, if the protocols used are previously published ones, citations should be given. Please revise this section to include this information.

2. Please ensure that you refer to Figure 7 in your text as, if accepted, production will need this reference to link the reader to the figure.

<h3>**3. Please upload a copy of Figure 6, to which you refer in your text. If the figure is no longer to be included as part of the submission please remove all reference to it within the text.**</h3>

4. We note that [Figure(s) 1-7] in your submission contain [map/satellite] images which may be copyrighted. All PLOS content is published under the Creative Commons Attribution License (CC BY 4.0), which means that the manuscript, images, and Supporting Information files will be freely available online, and any third party is permitted to access, download, copy, distribute, and use these materials in any way, even commercially, with proper attribution. For these reasons, we cannot publish previously copyrighted maps or satellite images created using proprietary data, such as Google software (Google Maps, Street View, and Earth). For more information, see our copyright guidelines: http://journals.plos.org/plosone/s/licenses-and-copyright.

1.    You may seek permission from the original copyright holder of Figure(s) [1-7] to publish the content specifically under the CC BY 4.0 license. 

Additional Editor Comments (if provided):

The submitted article is interesting and suitable for PlosOne, mainly due to the application of the APCS/MLR model. However, I have some concerns about this manuscript that should be reviewed:

1. Arsenic was analyzed, but As is not a heavy metal. In my opinion, "heavy metal" should be change by "Potentially Toxic Elements". Besides, it should be interesting to explain why these elements were analyzed (introduction or mat and methods). For example, Ba or Mn are urban contaminants due to their origin from industrial and traffic sources (Ba). Even Fe is an interesting tracer for geochemical assessment. I understand that As and Hg are related to industrial activities. Maybe it should be better to try to improve why these elements were selected.

2. Information about soil origin is missing. I saw that are several samples and it is impossible to write in a table. However, it should be interesting to try to write a short sentence about the origin of the sample. E.g. XX from parks and gardens, XX near to roads, XX industrial areas, XX agricultural areas. See for example https://doi.org/10.1007/s11368-017-1750-0
https://doi.org/10.1007/s11368-015-1304-2
https://doi.org/10.1007/s12665-019-8762-6

3. The Source apportionment method is very interesting and gives an alternative approach to typical papers on urban contamination. However, in my opinion, it should be well explained the APCS/MLR and PMF. A short explanation about why this method is more suitable than typical approaches (e.g. advantages) and a short explanation for readers to try to replicate for their works. In a previous work https://doi.org/10.1016/j.envpol.2018.09.147 the authors give more details, and it could be interesting also for this paper. If authors do not want to add into the manuscript, it could be interesting to add as supplementary data like this paper https://doi.org/10.1007/s00244-018-0572-4

4. Why was selected Ordinary kriging and not Inverse distance weighting (IDW) or SPLINE?

5. Line 203 "A higher SOM in some soils could be related to high application rates of organic fertilizers". First, this type of sentences should be written as discussion and soil properties are missing in the discussion section. In any case, the manuscript is missing some information about soil origin. I think that these higher values can be related to parks and gardens.

Besides, it should be interesting to try to see if contents of Cd, Cr or Cu are related to this SOM contents (Line 328). Sometimes are also related to garden-tending processes (protectors or wooden fences treated with chromate) as the most probable origin. Besides, P and N contents can help to explain well this relation between SOM and organic fertilizers https://doi.org/10.1007/s11368-015-1304-2

6. why other soils properties (e.g. P, N, soil texture, etc.) were not measured? In addition, how was measured pH and OM? (Calcination, Walkey black, etc)

7. I suggest a short paragraph with a comparison of studied soils with soils from other Chinese cities.

8. I think that references are not according to the journal guidelines. E.g. Journal names should be abbreviated https://journals.plos.org/plosone/s/submission-guidelines#loc-references

Minor comments

1. Line 38. Typo error.

Line 181. "eight" instead "8". Please, try to write the full name for numbers 1-10 when it's possible.

2. L126. Why 0.074 nm mesh?

3. Line 135. Which means the reference #23? Is it related to the methodological method?

4. Line 200. Please, also give for the pH values the max and min such as SOM contents.

5. Table 2 it should be after Line 225

Reviewers' comments:

Reviewer's Responses to Questions

**Comments to the Author**

1. Is the manuscript technically sound, and do the data support the conclusions?

Reviewer #1: Partly

Reviewer #2: Yes

2. Has the statistical analysis been performed appropriately and rigorously? 

Reviewer #1: Yes

Reviewer #2: Yes

3. Have the authors made all data underlying the findings in their manuscript fully available?

Reviewer #1: Yes

Reviewer #2: No

4. Is the manuscript presented in an intelligible fashion and written in standard English?

Reviewer #1: No

Reviewer #2: Yes

5. Review Comments to the Author

Reviewer #1: The manuscript entitled “Source apportionment of heavy metals in soils using APCS/MLR, PMF and geostatistics in a typical industrial and mining city in Eastern China” submitted to PONE, authors have determined source apportionment using multivariate techniques. I have few suggestions regarding this manuscript:

The authors should rewrite the abstract as: First explain the background of your work, objectives, then methods employed and main results of this study and finally what are the recommendations of this work to the society/enterprises/environmentalists.

The introduction section is very weak, authors should revise this section by following these articles:

Ecological Risk Assessment and Source Apportionment of heavy metal contamination in Agricultural Soils of North-eastern Iran.

Temporal distribution, source apportionment and pollution assessment of metals in the sediments of river Beas, India.

Assessment of heavy metal pollution in three different Indian water bodies by combination of multivariate analysis and water pollution indices.

Assessment of soil properties from catchment areas of Ravi and Beas Rivers: A review.

Pollution assessment of heavy metals in soils of India and ecological risk assessment: A State-of-the-Art.

Pollution assessment and spatial distribution of roadside agricultural soils: A case study from India.

Spatial distribution and potential ecological risk assessment of heavy metals in agricultural soils of Northeastern, Iran

Assessment of pollution in roadside soils by using multivariate statistical techniques and contamination indices.

Global evaluation of heavy metal content in surface water bodies: A meta-analysis using heavy metal pollution indices and multivariate statistical analyses.

Appraisal of metallic pollution and ecological risks in agricultural soils of Alborz province, Iran.

Ecological and human health risks appraisal of metal(loid)s in agricultural soils: a review.

Before objective, author should explain the rational of this study.

Line 93, Materials and methods should be rewritten as material and methods.

In study area, authors have not added any reference, from where this information was taken. Author should add references in this section. Why this area is important for this work. Explain it also.

In Table 1, authors have calculated C.V., S.D. and skewness, please discuss the following in the results and support your results by adding appropriate references.

In Table 2, the communality of the factor loadings missing, authors should add this in the Table 2.

Authors should add paragraph about the importance of this work to the society at the end of conclusion section.

Also improve the language of this manuscript.

Reviewer #2: Line 38 Print error.

Line 200 How did you determine pH and soil organic matter ?.

Line 204 Do not repeat data from table in text.

Table 1 should have median, percentile 25 and percentile 75.

Table 4. The Gaussian model should not test as it yields implausible results from its use (Oliver and Webster, 201).

Oliver, M.A., Webster, R., 2014. A tutorial guide to geostatistics: computing and modelling variograms and kriging. Catena 113, 56–69.

6. PLOS authors have the option to publish the peer review history of their article (what does this mean?). If published, this will include your full peer review and any attached files.

Reviewer #1: No

Reviewer #2: No

---

## [Author Response · Author response to Decision Letter 0]

6 May 2020

Dear editor and reviewers:

Thank you very much for your valuable comments. We have polished the manuscript and the revised manuscript has been submitted. Our revision have addressed all the concerns of the reviewers. The revised contents were highlighted in the manuscript with typeface and below is our point-by-point response. We look forward to your positive response. 

Best regards,

Authors: Cao Jianfei, Li Chunfang, Wu Quanyuan, Lv Jianshu

---

## [Decision Letter · Decision Letter 1]

1 Jul 2020

PONE-D-20-05425R1

Source apportionment of p otentially toxic elements in soils using APCS/MLR, PMF and geostatistics in a typical industrial and mining city in Eastern China

PLOS ONE

Dear Dr. Jianfei,

Thank you for submitting your manuscript to PLOS ONE. First, I want to apologize for the delay in reviewing this paper. Following this message are the reviews of the above-referenced manuscript. After reviewer's comments, some minor revisions are required.

We'll be glad to consider this paper for publication after it's been revised in accordance with the reviewers' comments. Therefore, we invite you to submit a revised version of the manuscript that addresses the points raised during the review process.

We look forward to receiving your revised manuscript.

Kind regards,

Andrés Rodríguez-Seijo, PhD

Academic Editor

PLOS ONE

Reviewers' comments:

Reviewer's Responses to Questions

**Comments to the Author**

1. If the authors have adequately addressed your comments raised in a previous round of review and you feel that this manuscript is now acceptable for publication, you may indicate that here to bypass the “Comments to the Author” section, enter your conflict of interest statement in the “Confidential to Editor” section, and submit your "Accept" recommendation.

Reviewer #1: All comments have been addressed

Reviewer #2: All comments have been addressed

Reviewer #3: (No Response)

2. Is the manuscript technically sound, and do the data support the conclusions?

Reviewer #1: Yes

Reviewer #2: Yes

Reviewer #3: Yes

3. Has the statistical analysis been performed appropriately and rigorously? 

Reviewer #1: Yes

Reviewer #2: Yes

Reviewer #3: N/A

4. Have the authors made all data underlying the findings in their manuscript fully available?

Reviewer #1: Yes

Reviewer #2: Yes

Reviewer #3: Yes

5. Is the manuscript presented in an intelligible fashion and written in standard English?

Reviewer #1: (No Response)

Reviewer #2: Yes

Reviewer #3: Yes

6. Review Comments to the Author

Reviewer #1: Dear Editor Authors have addressed all my comments, so I recommend the publication of this manuscript to Plos One journal.

Reviewer #2: (No Response)

Reviewer #3: I attached my comments in the pdf of the ms. Probably you can see my name there. Is no Problem for me. My main advise is that you add a table or a graph in which you compare your results of element concnetrations with those of other cities worldwide, that the meaning of the Longkou results can be classified.

7. PLOS authors have the option to publish the peer review history of their article (what does this mean?). If published, this will include your full peer review and any attached files.

Reviewer #1: No

Reviewer #2: No

Reviewer #3: No

---

## [Author Response · Author response to Decision Letter 1]

19 Jul 2020

Dear Reviewer:

Thank you very much for your valuable comments. We have polished the manuscript and the revised manuscript has been submitted. We look forward to your positive response.

Best regards,

Authors

---

## [Editor Report · Decision Letter 2]

19 Aug 2020

Source apportionment of p otentially toxic elements in soils using APCS/MLR, PMF and geostatistics in a typical industrial and mining city in Eastern China

PONE-D-20-05425R2

Dear Dr. Jianfei,

We’re pleased to inform you that your manuscript has been judged scientifically suitable for publication and will be formally accepted for publication once it meets all outstanding technical requirements.

Kind regards,

Andrés Rodríguez-Seijo, PhD

Academic Editor

PLOS ONE
---

## [Editor Report · Acceptance letter]

21 Aug 2020

PONE-D-20-05425R2 

Source apportionment of potentially toxic elements in soils using APCS/MLR, PMF and geostatistics in a typical industrial and mining city in Eastern China 

Dear Dr. Jianfei:

I'm pleased to inform you that your manuscript has been deemed suitable for publication in PLOS ONE. Congratulations! Your manuscript is now with our production department. 

Kind regards, 

on behalf of

Dr. Andrés Rodríguez-Seijo 

Academic Editor

PLOS ONE